# Exploring Knowledge of Antibiotic Use, Resistance, and Stewardship Programs among Pharmacy Technicians Serving in Ambulatory Care Settings in Pakistan and the Implications

**DOI:** 10.3390/antibiotics11070921

**Published:** 2022-07-08

**Authors:** Zia Ul Mustafa, Marriam Nazir, Hafiza Kiran Majeed, Muhammad Salman, Khezar Hayat, Amer Hayat Khan, Johanna C. Meyer, Brian Godman

**Affiliations:** 1Discipline of Clinical Pharmacy, School of Pharmaceutical Sciences, Universiti Sains Malaysia, Gelugor 11800, Penang, Malaysia; dramer2006@gmail.com; 2Department of Pharmacy Services, District Headquarter (DHQ) Hospital, Pakpattan 57400, Pakistan; 3Department of Medicine, Faisalabad Medical University, Faisalabad 38000, Pakistan; marriamnazir72@yahoo.com; 4Rural Health Center, Government of the Punjab, Faisalabad 38000, Pakistan; onlykh@icloud.com; 5Department of Pharmacy, The University of Lahore, Lahore 54700, Pakistan; msk5012@gmail.com; 6Institute of Pharmaceutical Sciences, University of Veterinary and Animal Sciences, Lahore 54000, Pakistan; khezar.hayat@uvas.edu.pk; 7Department of Public Health Pharmacy and Management, School of Pharmacy, Sefako Makgatho Health Sciences University, Ga-Rankuwa 0204, South Africa; hannelie.meyer@smu.ac.za; 8Centre of Medical and Bio-Allied Health Sciences Research, Ajman University, Ajman P.O. Box 346, United Arab Emirates; 9Strathclyde Institute of Pharmacy and Biomedical Science (SIPBS), University of Strathclyde, Glasgow G4 0RE, UK

**Keywords:** awareness, antimicrobials, antimicrobial resistance, antimicrobial stewardship programs, pharmacy technicians, ambulatory healthcare, Pakistan

## Abstract

Antimicrobial resistance (AMR) is a leading global health threat, increasing morbidity, mortality, and costs, with excessive and irrational use of antimicrobials contributing to the development of AMR. Consequently, the aims of this study were to evaluate the understanding of antibiotic use, AMR, and antimicrobial stewardship programs (ASPs) among pharmacy technicians serving in ambulatory healthcare settings in Pakistan. A cross-sectional survey was conducted among pharmacy technicians serving in 144 ambulatory care settings in seven districts of Punjab province using a validated questionnaire. Overall, 376 technicians completed the survey (85.8% response rate). The majority were men (89.1%), aged 25–35 years (45.1%), serving in emergency departments (43.9%) and filling 31–60 prescriptions per day (37.5%). Most (79.5%) knew that antibiotics were one of the most frequently prescribed drug classes, while 59.8% believed antibiotics for common colds did not speed up recovery. Inadequate duration (59.6%) and inadequate dosages (57.7%) of antibiotic therapy were reported as the leading causes of AMR. Terms including ‘superbugs’, ‘multidrug resistance’, and ‘extensively drug resistance’ were known to 42.0%, 25.3%, and 20.7% of participants, respectively; however, <10% knew about ASPs, including their core elements and purpose. Our study revealed that pharmacy technicians have adequate awareness of antibiotic use but are currently unaware of AMR and ASPs, which is a concern.

## 1. Introduction

Antimicrobial resistance (AMR) is a considerable and growing threat to global health, food security, and development [1,2]. Overall, AMR is associated with increased morbidity and mortality as well as increased length of hospital stay and overall costs [3,4,5,6,7,8]. In 2019, an estimated 4.95 million deaths globally were associated with AMR, with the greatest burden in sub-Saharan Africa and Asia [9]. If not addressed, gross domestic product (GDP) per country could be reduced by 3.8% by 2050 [10].

The extensive and inappropriate use of antibiotics is one of the leading causes of the development of AMR among disease-causing pathogens [2,11]. Irrational prescribing patterns, lack of standard diagnostic facilities, inappropriate awareness of AMR, insufficient infection control and preventive practices, and the lack of treatment guidelines are seen as prominent factors in the development of AMR [12,13,14,15]. Hospital settings, particularly in low- to middle-income countries (LMICs), are highly vulnerable to AMR due to excessive use of antibiotics where at least 20–50% of antibiotic utilization is reported to be irrational [2,13,16,17].

Pakistan is a LMIC in South Asia with high AMR rates enhanced by high consumption of antimicrobials [18,19]. Studies indicate 91% of *E. coli*, 91.7% of *S. Typhi*, 90.9% of *Acinetobacter* species, and 100% of *Klebsiella* species were reported to be resistant to amikacin, fluoroquinolone, imipenem, and third generation cephalosporins, respectively [20,21,22]. Emergence of extensively drug-resistant (XDR) typhoid in 2016 in Sindh Provinces, Pakistan, also led to appreciable global health concerns [23,24].

Health care workers (HCWs), including pharmacists and pharmacy technicians, play a critical role in educating patients about the appropriate use of antibiotics in ambulatory care and hospitals across countries. Other activities include minimizing the spread of infection through education about hygiene and personal protection, as well as through administering vaccines [25,26,27,28,29,30,31]. In addition, they are directly involved in the prescribing, dispensing, administration, and monitoring of antibiotic use in the complex medication use process in hospital settings across countries [17,32,33,34,35]. Hospital-based antimicrobial stewardship programs (ASPs) are considered an important tool to address the ongoing threat of AMR [2,17,35,36,37,38]. This is because ASPs aim to optimize the prescribing of antimicrobials by promoting rational antibiotic use, thereby reducing health care costs associated with AMR and its impact [2,17,39,40]. However, there are currently concerns with the implementation of ASPs in Pakistan [38].

Pakistan launched its national action plan (NAP) against AMR in 2017. Core objectives included the development and implementation of a national awareness strategy against AMR, the establishment of national surveillance of AMR and the enforcement of infection control and preventive practices [40,41]. In 2018, an assessment of ongoing activities stressed the need to establish a multisectoral approach to improve laboratory standards, enhance rational antibiotic use, as well as increase the awareness of antibiotics among HCWs and the general public [42]. This built on identified challenges with implementing the NAP [19].

We are aware that previous studies have been undertaken among HCWs in the different regions of Pakistan, including physicians and pharmacists as well as medical and pharmacy students, to ascertain key aspects regarding antibiotics use, AMR, and ASPs [38,43,44,45,46,47,48,49]. However, we are unaware of any studies that have been conducted to fully evaluate knowledge of antibiotic use, AMR, and ASPs among pharmacy technicians serving specifically in ambulatory care facilities. This is important since pharmacy technicians are involved in the procurement, storage, dispensing of prescriptions written by physicians, as well as the administration of medicines including antibiotics in health facilities treating both inpatients and ambulatory care patients. However, there have been studies assessing key factors associated with the inappropriate dispensing of antibiotics among non-pharmacy workers in Pakistan [50].

Consequently, we sought to address this by conducting a cross-sectional survey among pharmacy technicians in ambulatory care facilities in the Punjab province of Pakistan to provide guidance on future potential activities where there are concerns. This is because pharmacy technicians can play an effective role in combating AMR by ensuring appropriate dispensing of antibiotics against a valid prescription, manage used antibiotics, promote AMR awareness, counselling patients to complete their antibiotic regime, and cooperating with other healthcare professionals [51]. Moreover, pharmacy personnel can establish effective ASPs in ambulatory care settings [51,52,53]. There are also high rates of antibiotic dispensing without a prescription in Pakistan despite current regulations [54,55]. This also needs to be addressed going forward.

## 2. Results

We will first present the demographic characteristics of the study population before presenting their understanding of antibiotic use, AMR, including awareness and terminologies, and ASPs.

### 2.1. Demographic Characteristics of the Study Population

The investigators approached 438 pharmacy technicians conveniently sampled, of whom 376 subsequently participated in the survey, giving a response rate of 85.8%. The majority of the study participants were male (89.1%) and aged between 25–35 years (45.1%) (Table 1). They also mainly resided in rural areas (51.1%) and typically served as junior pharmacy technicians (56.1%). Most pharmacy technicians (43.9%) worked in the emergency department, filling 31–60 prescriptions per day (37.5%). Almost two thirds of the participants (63.6%) had not received any training related to antibiotic use. As far as the working experience of the study population is concerned, 26.3% had >10–15 years of working experience followed by 25.0% with 1–5 years of experience.

### 2.2. Understanding about Antibiotic Use

The majority of the study participants (79.5%) gave a correct response to the question that antibiotics were one of the frequently prescribed classes of drugs in Pakistan, while 59.8% considered that antibiotics used in patients with common colds do not contribute to a quicker recovery. Just over half (55.1%) of the pharmacy technicians surveyed believed antibiotics should not be prescribed for future bouts of influenza and 52.4% correctly stated that antibiotics should not be used to treat influenza. Just over two thirds of the study population (67.3%) answered correctly that antibiotics should not be stopped when symptoms improved. Of concern is that the vast majority of participants (93.9%) indicated that antibiotics can currently be obtained without a prescription in Pakistan. In addition, 36.2% believed antibiotics were first-line therapy for a sore throat and 43.1% believed left-over antibiotics can be used again in future for a similar infection (Table 2). The mean and median scores of antibiotic use among the study participants were 8.045 and 8.000 respectively, with two thirds of the study population (65.2%) possessing good knowledge of antibiotics, while 34.8% had poor knowledge (Figure 1).

### 2.3. Understanding about Antibiotic Resistance

More than three quarters of the participants (76.1%) stated that HCWs are responsible for the transmission of resistant strains from an infected to a healthy person. However, 45.2% believed resistant strains cannot spread in healthcare institutions. Inadequate duration and inadequate dosage of antibiotic therapy are the leading causes of antibiotic resistance according to 59.6% and 57.7% of the participants, respectively (Table 3). However, only 24.7% knew that shifting from empiric to targeted antibiotic therapy after culture and sensitivity testing can help minimize AMR. The mean and median antibiotic resistance score of the pharmacy technicians were 4.079 and 4.000, respectively. Overall, 60.4% had good knowledge whereas 39.6% presented with poor knowledge of antibiotic resistance (Figure 1).

### 2.4. Awareness about the Terminologies Related to Antibiotic Resistance

Awareness of the different terminologies regarding AMR are given in Table 4. Most technicians knew about the terms ‘AMR’ (81.4%), ‘antibiotic resistance’ (91.8%), and ‘drug resistance’ (76.9%). However, specific terms including ‘superbugs’, ‘multidrug resistance’, and ‘extensive drug resistance’ were only known among a minority. The majority of pharmacy technicians (80.9%) were also unaware of the ‘National Action Plan of Pakistan against AMR’. The mean and median score about awareness of terminologies related to antibiotic resistance were 5.156 and 5.000, respectively. Less than one third of those surveyed (28.7%) possessed good knowledge of terminologies related to antibiotic resistance (Figure 1).

### 2.5. Awareness of the Antimicrobial Stewardship Program (ASP)

Only 7.4% of pharmacy technicians surveyed knew about the term ‘antimicrobial stewardship program’ and only 6.4% knew about the components of ASP (Table 5). A total of 7.4% and 8.5% pharmacy technicians respectively knew that ASPs can reduce the cost of antimicrobials and improve patients’ quality of life. The mean score of study participants concerning awareness of ASPs was 0.558. Only 14.4% pharmacy technicians had good knowledge of ASPs (Figure 1).

### 2.6. Association of Understanding about Antibiotics with Demographic Characteristics

The association between demographic characteristics of pharmacy technicians and antibiotic use (ABU), AMR, antibiotic terminologies (ABT), and ASP with demographic variables is documented in Table 6. A statistically significant association was identified between female, junior pharmacy technicians, those who had any training and those with 1–5 years’ experience, and their awareness about antibiotic use, AMR, and terminologies related to AMR (*p* < 0.05).

### 2.7. Sources of Information

Social media, including Facebook and WhatsApp, were the most common sources of information regarding antimicrobials for the majority of pharmacy technicians surveyed (41.0%). Other sources of information included healthcare professionals (34.6%) and brochures/leaflets (13.0%) available with the drug packaging (Figure 2).

## 3. Discussion

This study provided an in-depth understanding of pharmacy technicians’ knowledge regarding antibiotics and AMR in Pakistan. The findings showed that most pharmacy technicians had a reasonable understanding of antibiotics and their use, with approximately two thirds being aware that antibiotics are frequently prescribed, there can be allergies and patients should not discontinue antibiotics when symptoms are improving. These findings are similar to the previous study conducted in Pakistan among pharmacy students [47]. However, of concern was a poor understanding regarding the preventive use of antibiotics for future infections, antibiotics can be effective for viral infections, left-over antibiotics can be re-used, and the availability of antibiotics without a prescription. These findings, though, are better than those in a study from India where less than a quarter of Bachelor of Pharmacy students knew about the use of antibiotics for common colds [56].

Encouragingly, our study reported that excessive use of antibiotics, inappropriate dosages, and duration, as well as a lack of regulatory enforcement that permit easy availability of antibiotics, are common factors resulting in irrational use of antibiotics and, consequently, rising AMR rates in Pakistan. These findings are similar to the previous study in India where most paramedic staff knew that the overuse of antibiotics was a leading cause of AMR [57]. However, most pharmacy technicians were unaware of terminology such as superbugs, multidrug resistant and extensive drug resistance infection, and the National Action Plan of Pakistan against AMR. This is a concern, similar to a study from Sri Lanka where most pharmacy students were unaware about terms such as superbugs [58].

Our study also revealed that most pharmacy technicians were not aware of the core elements of ASPs, the purpose of ASP, and how ASPs enhance the appropriate utilized antimicrobial agents. The primary reasons for this could be the lack of these initiatives in Pakistan, specifically at these remote health facilities. The lack of knowledge in these areas was also related to status and experience, which needs to be factored in going forward. These findings though are similar to studies in other LMICs where health professionals, including pharmacists, had inadequate knowledge of antibiotics and ASPs [59,60,61]. This again needs addressing going forward in Pakistan. Potential activities to address these concerns include instigating appropriate educational activities among pharmacy technicians during their diploma training and post qualification with the help of senior technicians and pharmacists in the various settings [2,62]. This is particularly important since social media, including Facebook and WhatsApp, were the most common sources of information for the surveyed pharmacy technicians rather than other healthcare professionals or any national guidelines which should be updated as part of national action plan activities [19]. Any guidelines produced should be robust, easy to follow, and regularly updated to enhance their use and improve future antimicrobial prescribing [63,64]. Educational activities can also assist with addressing some of the misinformation that can arise from social media, which was particularly prevalent during the early stages of the COVID-19 pandemic. This included the widespread promotion of re-purposed treatments including hydroxychloroquine, remdesivir, and ivermectin despite limited evidence, all of which were subsequently shown to have limited benefit alongside harm in some patients [65,66,67,68,69,70,71].

We are aware of a number of limitations with our study. Firstly, we collected data only from seven districts of Punjab. Consequently, we are unable to fully generalize the findings of our study to the rest of country. Moreover, we employed a convenient sampling technique that can be associated with bias. However, in view of the characteristics of the Punjab region in Pakistan, coupled with our methodology, we believe our findings are robust for Pakistan and can be helpful with improving the appropriate use of antibiotics within ambulatory care facilities in Pakistan. We will be looking to take this research forward in future studies.

## 4. Materials and Methods

### 4.1. Study Setting and Design

The Department of Health of the Punjab government has two divisions, of which the first is the Specialized Health and Medical Education Department, that is the controlling authority of tertiary/teaching hospitals, mainly established in metropolitan cities of the province and serve as referral hospitals. The second division is the Ambulatory and Secondary Healthcare Department, comprising of secondary hospitals, including district headquarters (DHQs), tehsil headquarters (THQs), ambulatory care health settings including rural health centers (RHCs), and basic health units (BHUs).

There are currently 358 RHCs in Punjab, each comprising approximately 20–25 beds, operating 24/7, and serving around a 100,000 nearby population. More than 2857 BHUs currently provide services in the province, with each Union Council having at least one BHU, with 2–5 beds, serving 10,000 to 25,000 people [72]. All BHUs and RHCs have ambulatory care physicians, nurses, and other technicians, including pharmacy technicians as staff members. Free medical checkups, medicines, and laboratory facilities are provided in these ambulatory healthcare facilities. Pharmacists are employed in tertiary, DHQ, and THQ hospitals, while pharmacy technicians are currently serving at ambulatory health care facilities. These pharmacy technicians have a one-year educational diploma training, followed by further training under the supervision of a pharmacist or senior pharmacy technician after matriculation. Pharmacy technicians are responsible for the storage, distribution, dispensing, and administration of medicines.

We used a cross-sectional study design to collect data regarding knowledge of antibiotic use, AMR, and ASPs among pharmacy technicians, serving in ambulatory care facilities in the public sector of Punjab, over a two-month period (Feb–March 2022).

### 4.2. Study Sites

We collected data from pharmacy technicians from 37 RHCs and 107 BHUs from seven districts of Punjab out of 36 districts that were conveniently sampled. There were 2–3 pharmacy technicians at each BHU and 4–5 at each RHC.

### 4.3. Study Sample

The sample size was calculated using the Raosoft, 206-525-4025 (US), online sample size calculator [73]. Assuming an expected frequency of 50%, at a 95% confidence interval, and 5% margin of error, the minimum sample size was calculated at 376.

Pharmacy technicians currently serving in the ambulatory health care department, government of Punjab, belonging to the seven districts (Sahiwal, Pakpattan, Okara, Qasor, Vehari, Bahawalnagar, and Multan) were included in the survey. As mentioned, these districts were selected using a convenient sampling technique. Pharmacy technicians working in the private sector, pharmacy technician students, and those working in other districts of Punjab were excluded from the current study as we wanted to concentrate initially in the public sector.

### 4.4. Development of the Study Instrument

The study tool used in our survey was adapted from previous studies conducted in the local population [47,48] as well as a review of the literature. The initial draft of the study instrument was prepared by the leading investigators (ZUM and MS). Pilot testing was subsequently conducted among 20 pharmacy technicians from eight ambulatory health care facilities of two districts to verify the clarity and understanding of the content, to enhance its robustness. Cronbach’s alpha was calculated based on the results of the pre-testing of the questionnaire in order to determine its reliability. The value of Cronbach’s alpha was higher than seven, which indicates an acceptable level of internal consistency. The participants from the pilot study were not included in the principal study.

Based on the outcome of the pilot study, minor amendments were made to the questionnaire. This resulted in a final study instrument comprising five sections. Appendix A covered the demographic characteristics of the study population. Ten questions were included in this section incorporating age, city, sex, marital status, residence, designation, status of training, working department, years of experience, and the number of prescriptions filled per day. Appendix A consisted of nine questions, including questions regarding perspectives of antibiotic use among study participants. Each question had three options, namely ‘Yes’, ‘No’, and ‘Do not know’. Participants were requested to select one option out of the three based on their knowledge. Appendix A contained eight questions about participants’ understanding of AMR. Pharmacy technicians were asked to choose one correct option from ‘Yes’, ‘No’, and ‘Unsure’. Appendix A covered awareness of different terminologies related to AMR, including antibiotic resistance, superbugs, and extensively drug resistance infections. This section consisted of nine questions and participants were asked to choose ‘Yes’ or ‘No’ according to their awareness about certain terminology. Appendix A contained five questions about knowledge of ASPs, requiring participants to select one option from ‘Yes’, ‘No’, and ‘Unsure’. The last question asked participants about their source/s of information about infections and antibiotics, including social media, health care professionals, and leaflets/brochures included with the antibiotic packs dispensed.

### 4.5. Data Collection Procedures

The data collection tool was distributed among pharmacy technicians during duty hours (8:00 to 16:00) with a request to complete the questionnaire in a timely manner. The pharmacy technicians were provided with study instruments, and these were collected after half an hour in a sealed box. Participation in the study was entirely voluntary and anonymous. Potential study participants were firstly briefed about the purpose of the study prior to their enrollment. Written informed consent was obtained from all participants, and anyone could leave the survey at any time of data collection.

### 4.6. Statistical Analysis

Descriptive statistics, such as frequencies and percentages, were used to analyze the data. Moreover, the normality of the data was checked through Shapiro–Wilks and Kolmogorov–Smirnov tests. One mark was allocated for each correct response against the understanding of respondents towards different aspects of antibiotics, with no marks given for incorrect or unsure responses. Based on the median score, the outcomes regarding the understanding of respondents towards ABU (antibiotic use), ABR (antibiotic resistance), ABT (antibiotic terminologies), ASP (antimicrobial stewardship program) were dichotomized as “Good” versus “Poor” (Table 7). The median score was calculated from the sum of the total score of the respondents in each domain, similar to the methodology used by Horvat et al. (2017) [74].

Independent Samples Mann–Whitney U Test and Independent Samples Kruskal–Wallis Test were performed to test if there were differences among characteristics of the pharmacy technicians with regard to their attitudes, perceptions, willingness, and motivation towards PBR. All data analysis was performed using the Statistical Package for the Social Sciences (SPSS Inc., version 18, IBM, Chicago, IL, USA). *p*-values < 0.05 were considered statistically significant.

## 5. Conclusions and Recommendations

Overall, our study showed that pharmacy technicians serving in ambulatory care settings in the Punjab province in Pakistan had a good understanding of antibiotic use and resistance. However, there were concerns including certain terminologies and awareness of the NAP of Pakistan against AMR. Moreover, there was limited knowledge regarding ASPs. Consequently, recommendations going forward include the following:The current curriculum of pharmacy technicians must be updated to include subjects related to antibiotic use, AMR, and ASPs including the future role of pharmacy technicians in helping to instigate ASPs in ambulatory care and monitor subsequent antibiotic utilizationTraining and awareness sessions must be instigated periodically for pharmacy technicians to ensure appropriate use and dispensing of antibiotics in ambulatory care facilities. This can be part of future continual professional development activitiesPharmacists need to be appointed at ambulatory care facilities alongside pharmacy technicians to improve future dispensing of antibiotics given current concernsEstablishing ASPs in ambulatory care settings is a must for all health care providers, including pharmacists and pharmacy technicians, given the current extent of antimicrobial utilization in ambulatory care made worse by the recent COVID-19 pandemic.

## Figures and Tables

**Figure 1 antibiotics-11-00921-f001:**
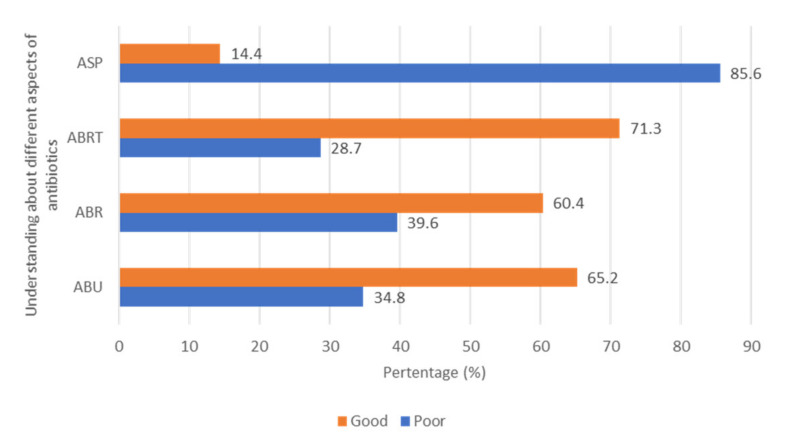
Understanding of respondents regarding ABU (antibiotic use), ABR (antibiotic resistance), ABRTs (antibiotic resistance terminologies), ASP (antimicrobial stewardship program).

**Figure 2 antibiotics-11-00921-f002:**
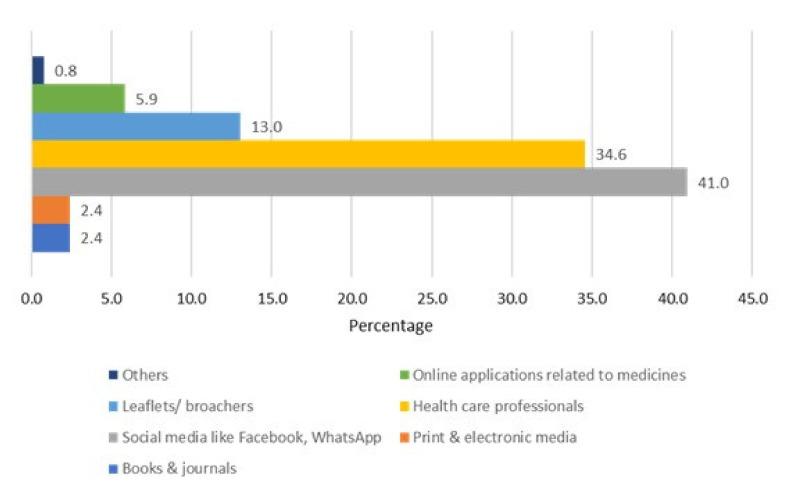
Pharmacy technicians’ sources of information on antibiotic use.

**Table 1 antibiotics-11-00921-t001:** Demographic characteristics of the pharmacy technicians (n = 376).

Variables	Frequency (n)	Percentage (%)
**Sex**		
Female	41	10.9
Male	335	89.1
**Age**		
<25 years	17	4.5
25–35 years	170	45.1
36–46 years	147	39.0
>46 years	43	11.4
**Residence**		
Rural	192	51.1
Urban	184	48.9
**Designation**		
Junior	211	56.1
Senior	165	43.9
**Training**		
Yes	137	36.4
No	239	63.6
**Department**		
Emergency	165	43.9
Indoor	81	21.5
Outdoor	130	34.6
**Experience**		
1–5 years	94	25.0
>5–10	93	24.7
>10–15	99	26.3
>15–20	59	15.7
>20	31	8.2
**City (Districts)**		
Bahawalnagar	51	13.5
Multan	74	19.6
Okara	42	11.1
Pakpattan	41	10.9
Qasor	31	8.2
Sahiwal	76	20.2
Vehari	61	16.2
**Prescriptions filled per day**		
1–30	96	25.5
31–60	141	37.5
61–100	83	22.1
101–150	42	11.2
>150	14	3.7

**Table 2 antibiotics-11-00921-t002:** Understanding of pharmacy technicians about antibiotic use (n = 376).

Questions	Yes (%)	No (%)	Unsure (%)
Antibiotics are the most commonly prescribed anti-infective agents by public healthcare sector facilities	299 (79.5)	28 (7.4)	49 (13.0)
Common colds treated with antibiotics will make patients recover more quickly	85 (22.6)	225 (59.8)	66 (17.6)
Antibiotics should be prescribed as preventive measures to fight against future microbial attacks	118 (31.4)	207 (55.1)	51 (13.6)
Antibiotics cannot treat influenza	108 (28.7)	197 (52.4)	71 (18.9)
Antibiotics are indicated to relieve pain	41 (10.9)	331 (88.0)	4 (1.1)
Antibiotics might develop allergies in susceptible individuals	289 (76.9)	31 (8.2)	56 (14.9)
Diphenhydramine is an antibiotic used in treating upper respiratory tract infections	58 (15.4)	255 (67.8)	63 (16.8)
Cefotaxime belongs to the third-generation cephalosporins	249 (66.2)	79 (21.0)	48 (12.8)
Patients can stop taking antibiotics when symptoms improve	81 (21.5)	253 (67.3)	42 (11.2)
It is good practice to keep antibiotics that are left over from a prescribed course for the next time treatment is needed for the same type of infection	191 (50.8)	162 (43.1)	23 (6.1)
Antibiotics treatment can eliminate most of the sensitive bacterial cells from patients	95 (25.3)	78 (20.7)	203 (54.0)
Antibiotics can be obtained without a prescription in Pakistan	353 (93.9)	21 (5.6)	2 (0.5)
Antibiotics are the first line of treatment for a sore throat	136 (36.2)	199 (52.9)	41 (10.9)
Mean	8.045		
Median	8.000		
Std. Deviation	1.930		

**Table 3 antibiotics-11-00921-t003:** Understanding of pharmacy technicians about antibiotic resistance (n = 376).

Questions	Yes (%)	No (%)	Unsure (%)
A resistant bacterium cannot spread in healthcare institutions	170 (45.2)	145 (38.6)	61 (16.2)
Healthcare workers serve as vectors carrying resistant strains from infected patients to healthy patients	286 (76.1)	45 (12.0)	45 (12.0)
Exposure to antibiotics appears to be the main risk factor for the emergence of antibiotic resistant bacteria.	234 (62.2)	67 (17.8)	75 (19.9)
The inadequate duration of therapy contributes to antibiotic resistance leading to poor patient compliance	224 (59.6)	63 (16.8)	89 (23.7)
Inadequate dosages contribute to antibiotic resistance due to poorly designed dosing regimens	217 (57.7)	60 (16.0)	99 (26.3)
Antimicrobial resistance can be minimized by changing empiric therapy to a selected narrow-spectrum therapy in response to the availability of culture and sensitivity results.	93 (24.7)	56 (14.9)	227 (60.4)
Cross resistance is the condition in which resistance occurs to a particular antibiotic that often results in resistance to other antibiotics, usually from a similar class	84 (22.3)	48 (12.8)	244 (64.9)
Lack of enforcement of regulations sometimes permits antibiotics to be purchased without a prescription from pharmacies	251 (66.8)	32 (8.5)	93 (24.7)
Mean	4.079		
Median	4.000		
Std. Deviation	1.734		

**Table 4 antibiotics-11-00921-t004:** Awareness about the terminologies related to antibiotic resistance.

Questions	Yes (%)	No (%)
‘Antimicrobial resistance’?	306 (81.4)	70 (18.6)
‘Antibiotic resistance’?	345 (91.8)	31 (8.2)
‘Superbugs’?	158 (42.0)	218 (58.0)
‘Drug resistance’?	289 (76.9)	87 (23.1)
‘Multidrug resistance’?	95 (25.3)	281 (74.7)
‘Extensively drug resistance’?	78 (20.7)	298 (79.3)
‘Use of antibiotics with caution’?	300 (79.8)	76 (20.2)
‘Antibiotic resistance spreading very fast’?	296 (78.7)	80 (21.3)
‘National Action Plan for Antimicrobial Resistance in Pakistan’?	72 (19.1)	304 (80.9)
Mean	5.156	
Median	5.000	
Std. Deviation	1.545	

**Table 5 antibiotics-11-00921-t005:** Awareness about the antimicrobial stewardship programs (ASPs).

Questions	Yes (%)	No (%)	Unsure (%)
Heard about the term antimicrobial stewardship programs (ASPs)?	28 (7.4)	66 (17.6)	282 (75.0)
Knew about the core components of antimicrobial stewardship programs (ASPs)?	24 (6.4)	91 (24.2)	261 (69.4)
Knew that antimicrobial stewardship programs promote reasonable prescription of antimicrobials?	33 (8.8)	65 (17.3)	278 (73.9)
Antimicrobial stewardship programs help control antimicrobial resistance	35 (9.3)	82 (21.8)	259 (68.9)
Antimicrobial stewardship programs reduce the overuse of antimicrobials	30 (8.0)	105 (27.9)	241 (64.1)
Antimicrobial stewardship programs reduce the cost of treatment.	28 (7.4)	74 (19.7)	274 (72.9)
Antimicrobial stewardship programs improve medical quality.	32 (8.5)	105 (27.9)	239 (63.6)
Mean	0.558		
Median	0.000		
Std. Deviation	1.636		

**Table 6 antibiotics-11-00921-t006:** Test of statistical significance of variation in the pharmacy technicians’ understanding towards ABU, ABR, ABT, and ASP by their characteristics.

Variables	ABU		ABR		ABT		ASP	
	Mean Rank	*p*-Value	Mean Rank	*p*-Value	Mean Rank	*p*-Value	Mean Rank	*p*-Value
**Gender**								
Female	144.11	0.005	132.09	<0.001	154.71	0.033	185.00	0.758
Male	193.39		194.86		192.09		188.37	
**Age**								
<25 years	195.97	0.099	211.12	0.127	188.97	0.570	181.32	0.922
25–35 years	189.33		195.53		196.89		190.80	
36–46 years	176.67		172.84		180.76		187.15	
>46 years	222.74		205.45		181.79		186.92	
**Residence**								
Rural	178.92	0.077	179.25	0.087	177.67	0.043	186.80	0.611
Urban	198.50		198.15		199.80		190.28	
**Designation**								
Junior	147.90	<0.001	147.23	<0.001	160.82	<0.001	187.16	0.657
Senior	240.42		241.28		223.90		190.22	
**Training**								
Yes	214.37	<0.001	211.19	0.002	212.55	0.001	195.80	0.106
No	173.67		175.49		174.71		184.32	
**Department**								
Emergency	192.69	0.797	181.77	0.547	188.85	0.276	181.77	0.125
Indoor	185.85		195.72		203.07		187.65	
Outdoor	184.83		192.55		178.98		197.57	
**Experience**								
1–5 years	111.40	<0.001	114.18	<0.001	139.51	<0.001	187.88	0.936
5–10	164.14		159.89		168.39		192.38	
11–15	221.28		228.27		219.55		187.09	
16–20	249.01		254.66		227.30		183.58	
>20	275.52		246.76		224.42		192.61	
**Prescriptions filled/day**								
1–30	191.26	0.151	192.32	0.874	179.11	0.272	187.32	0.132
31–60	189.82		187.21		204.02		189.19	
61–100	166.39		179.56		183.19		179.58	
101–150	216.19		199.68		171.77		211.15	
>150	204.29		194.75		178.14		174.57	

NB: ABU = antibiotics use, ABR = antibiotic resistance, ABT = antibiotic terminologies, ASP = antimicrobial stewardship program.

**Table 7 antibiotics-11-00921-t007:** Scoring criteria for respondents regarding their understanding towards different aspects of antibiotics based on median scores.

	ABU	ABR	ABT	ASP
Poor	<8.000	<4.000	<5.000	<1.000
Good	≥8.000	≥4.000	≥5.000	≥1.000

## Data Availability

Available on reasonable request from the corresponding authors.

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
