# Peer review of "Exploring Knowledge of Antibiotic Use, Resistance, and Stewardship Programs among Pharmacy Technicians Serving in Ambulatory Care Settings in Pakistan and the Implications"

_antibiotics, 2022, doi:10.3390/antibiotics11070921_

Round 1
Reviewer 1 Report
Thank you for the opportunity to review the manuscript, describing the knowledge and awareness of antibiotic use, antimicrobial resistance, and antimicrobial stewardship among pharmacy technicians in Pakistan.
The focus of this survey is an important topic in the Asia region. The overall text is well written and the survey is well designed. However, the methods and results state what was slightly unclear and include several concerns. In addition, the results were not discussed sufficiently.
I have several comments that might improve this manuscript:
Specific comments
1) L88-94
The role of pharmacy technicians serving in ambulatory care facilities is unclear for readers in other countries. Please provide further detail including differences from general pharmacists in Pakistan and the knowledge gap.
2) Can/Do the pharmacy technicians in this study dispense antibiotics without a prescription?
If yes, their attitude would be affected increment of inappropriate antibiotic use.
3) Do marital status affect antibiotic awareness? Moreover, the result of this status is not discussed in this manuscript. This question may be inadequate in the viewpoints of diversity and human rights.
4) Discussion
Why are the pharmacy technicians that have a professional license insufficient understanding and awareness of antimicrobial stewardship? Please provide the author's specific insights from this viewpoint.
5) Table 6
The results shown in Table 6 were not discussed sufficiently. The authors should provide further detail on the differences in each characteristic.
6) Figure 1 and Table 7
Outcomes regarding the understanding of respondents were dichotomized as “Good” versus “Poor” based on each score. This comparison leads to natural results of "Poor" is poor understanding of each element of understanding aspects of antibiotics. I cannot understand what meaning of this analysis and results.
Author Response
Comments and Suggestions for Authors
Thank you for the opportunity to review the manuscript, describing the knowledge and awareness of antibiotic use, antimicrobial resistance, and antimicrobial stewardship among pharmacy technicians in Pakistan.
The focus of this survey is an important topic in the Asia region. The overall text is well written and the survey is well designed. However, the methods and results state what was slightly unclear and include several concerns. In addition, the results were not discussed sufficiently.
Author comments: Thank you for your positive comments. We hope we have now sufficiently addressed these.
I have several comments that might improve this manuscript:
Specific comments
1) L88-94
The role of pharmacy technicians serving in ambulatory care facilities is unclear for readers in other countries. Please provide further detail including differences from general pharmacists in Pakistan and the knowledge gap.
Author comments: Thank you for the comment. We have updated the manuscript and included the roles of pharmacy technicians at these health facilities. We hope this is now OK.
2) Can/Do the pharmacy technicians in this study dispense antibiotics without a prescription?
If yes, their attitude would be affected increment of inappropriate antibiotic use.
Author comments: Thank you for the comment. All pharmacy technicians are allowed to dispense antibiotics according to the prescription from registered Medical Practitioners. We have now inserted this into the revised paper.
3) Do marital status affect antibiotic awareness? Moreover, the result of this status is not discussed in this manuscript. This question may be inadequate in the viewpoints of diversity and human rights.
Author comments: Thank you – we agree and have now deleted this information from the relevant Tables and text, and hope this is now acceptable.
4) Discussion
Why are the pharmacy technicians that have a professional license insufficient understanding and awareness of antimicrobial stewardship? Please provide the author's specific insights from this viewpoint.
Author comments: Thank you for the comment. We have mentioned that there is lack of awareness of ASPs among pharmacy technicians because these initiatives were typically not taken into account at these health facilities, and have suggested potential ways forward. We hope this is now OK
5) Table 6
The results shown in Table 6 were not discussed sufficiently. The authors should provide further detail on the differences in each characteristic.
Author comments: Thank you for the comment. We tried to improve the result description of Table 6 along with the implications in the Discussion. We hope this is now acceptable.
6) Figure 1 and Table 7
Outcomes regarding the understanding of respondents were dichotomized as “Good” versus “Poor” based on each score. This comparison leads to natural results of "Poor" is poor understanding of each element of understanding aspects of antibiotics. I cannot understand what meaning of this analysis and results.
Author comments: Thank you for this comment. One mark was given for each correct response against understanding of respondents towards different aspects of antibiotics and no marks were given for no or unsure response. Based on median score (as indicated in Table 7), we grouped respondents into two groups such as ‘Good’ and ‘Poor’. This median score was calculated from the sum of the total score of respondents in each domain. Previously, researchers such as Horvat et al have utilized similar method as indicated in our study (reference now added in). We hope this is now OK.
Reviewer 2 Report
Your manuscript titled as 'Exploring knowledge of antibiotic use, resistance, and stewardship programmes among pharmacy technicians serving in ambulatory care settings in Pakistan and the implications' was interesting. However, there are some improvement parts in your manuscript. Please re-consider below points;
1) In the 'Results' part, you used the words "Most" and "Majority" (eg. Page 3, Lines 113, 115, 116, etc) But ”Most” and "majority" are misleading, for example, only about 51% of the respondents live in the rural area, which is almost the same as the percentage living in the city. Moreover, the same applies to the age groups: 45% of the 25-35 age group were "Most" as you said, but 39% of the 36-46 age group was not so different, so these two age groups are the "Most" group, I think.
2) I cannot understand the methodology of statistical analysis. You wrote "The mean and median scores" in the Lines 134, You mention the score, but there is no mention of it in the 'Materials and Methods'. How were the scores distributed? How many is the perfect score? Please describe in detail in the Methods section how the scoring was done. Also, the same can be said for the "2.6 Association of median scores with demographic characteristics" (Lines 187). You mention "median score." what does this mean? How to obtain the median score and what it means should also be detailed in the "Methods" section.
3) The analysis of ”2.7 Sources of information" (Lines 201) is too rough. Information about pharmacists as a whole may be important, but since your earlier analysis indicated that "inexperienced", gender, young pharmacists, etc. were statistically significant, it would be more telling to clarify what sources of information those characteristics are using.
4) There is minor point; The terms "Figure" and "Fig" are mixed (eg. Lines 136, 152, etc)
Author Response
Comments and Suggestions for Authors
Your manuscript titled as 'Exploring knowledge of antibiotic use, resistance, and stewardship programmes among pharmacy technicians serving in ambulatory care settings in Pakistan and the implications' was interesting. However, there are some improvement parts in your manuscript. Please re-consider below points.
Author comments: Thank you for your positive comments. We hope we have adequately addressed areas of concern.
1) In the 'Results' part, you used the words "Most" and "Majority" (eg. Page 3, Lines 113, 115, 116, etc). But ”Most” and "majority" are misleading, for example, only about 51% of the respondents live in the rural area, which is almost the same as the percentage living in the city. Moreover, the same applies to the age groups: 45% of the 25-35 age group were "Most" as you said, but 39% of the 36-46 age group was not so different, so these two age groups are the "Most" group, I think.
Author comments: Thank you for this comment. We have now revised this in the text, and hope this is now OK.
2) I cannot understand the methodology of statistical analysis. You wrote "The mean and median scores" in the Lines 134, You mention the score, but there is no mention of it in the 'Materials and Methods'. How were the scores distributed? How many is the perfect score? Please describe in detail in the Methods section how the scoring was done. Also, the same can be said for the "2.6 Association of median scores with demographic characteristics" (Lines 187). You mention "median score." what does this mean? How to obtain the median score and what it means should also be detailed in the "Methods" section.
Author comments: Thank you for this comment. As indicated in the revised paper, one mark was given for each correct response against understanding of respondents towards different aspects of antibiotics and no marks were given for incorrect or unsure response. Based on median score (as indicated in Table 7), we grouped respondents into two groups such as ‘Good’ and ‘Poor’. This median score was calculated from the sum of the total score of respondents in each domain. Previously, researchers such as Horvat et al have utilized similar method as indicated in our study (now referenced).
This information has been added in the revised version of the manuscript.
The title of 2.6 has also been revised as it was misleading. We utilized Independent Samples Mann-Whitney U Test and Independent Samples Kruskal-Wallis Test were performed to test if there were differences among characteristics of the pharmacy technicians with regard to their attitudes, perceptions, willingness and motivation towards PBR. Here, we reported mean rank with p-value instead of median score and IQR. The similar method as been utilized in previous studies. We hope this is now acceptable.
3) The analysis of ”2.7 Sources of information" (Lines 201) is too rough. Information about pharmacists as a whole may be important, but since your earlier analysis indicated that "inexperienced", gender, young pharmacists, etc. were statistically significant, it would be more telling to clarify what sources of information those characteristics are using.
Author comments: Thank you. We have highlighted in Figure 2 the main sources of information – and in the Discussion discussed appropriate ways forward to enhance appropriate antibiotic prescribing in the future. We hope this is now OK.
4) There is minor point; The terms "Figure" and "Fig" are mixed (eg. Lines 136, 152, etc)
Author comments: Thank you. We have corrected it.
Reviewer 3 Report
1.A well written paper and a welldone study! The study analysis and the data presentatio can be improved a lot. Please refer to the following points:
what is the meaning of pharmacy technicians who work in the emergency department? In the methods section, it says that they worked in the ambulatory clinics. It would be nice to describe their educational qualification (have they finished bachelor level studies or only training in pharmacy, etc?), their responsibilities to be able for international readers to comprehend the study population.
2. Writing prescriptions? Are the doctors predcribing the prescripions and they judt provide.
3. Provide the questionnaire as supplementary material and the scoring assigned.
4. It would be good to assess subsets of questionnaire instead of specific questions.
5. Reliability analysis of the questionnaire will be a valuable addition.
Author Response
Comments and Suggestions for Authors
A well written paper and a well-done study!
Author comments: Thank you for your positive comments – appreciated!
The study analysis and the data presentation can be improved a lot. Please refer to the following points:
Author comments: Thank you for these – we hope we have adequately addressed them
1. What is the meaning of pharmacy technicians who work in the emergency department? In the methods section, it says that they worked in the ambulatory clinics. It would be nice to describe their educational qualification (have they finished bachelor level studies or only training in pharmacy, etc?), their responsibilities to be able for international readers to comprehend the study population.
Author comments: Thank you for your comment. The educational qualification as well as their roles at these health facilities have been provided. We hope this is now OK.
2. Writing prescriptions? Are the doctors prescribing the prescriptions and they just provide.
Author comments: Thank you for the inquiry. The prescriptions are written by registered medical Practitioners and pharmacy technicians are filling them accordingly. We have updated the manuscript to reflect this, and hope this is now OK.
3. Provide the questionnaire as supplementary material and the scoring assigned.
Author comments: Thank you – now inserted
4. It would be good to assess subsets of questionnaire instead of specific questions.
Author comments: Thank you for the comment. We have analyzed this both ways. Each question has been analyzed individually and the frequency and percentage written in each subset (Table 2, 3, 4, 5) as well as in the subset form (Table 6). We hope this is OK with you
5. Reliability analysis of the questionnaire will be a valuable addition
Author comments: Cronbach’s alpha was calculated based on the results of the pre-testing of the questionnaire, to determine its reliability. The value of Cronbach’s alpha was higher than seven, indicating an acceptable level of internal consistency. This information has now been incorporated into the manuscript with minor changes to indicate where we have included this data. We hope this is now acceptable.
Round 2
Reviewer 2 Report
The authors correctly understood the intent of the question and the paper has been improved. I believe it deserves acceptance.
Reviewer 3 Report
All comments addressed! Its fine!